# Relationship between Step-by-Step Foot Kinematics and Sprint Performance

**DOI:** 10.3390/ijerph19116786

**Published:** 2022-06-01

**Authors:** Isabel Martín-Fuentes, Roland van den Tillaar

**Affiliations:** 1Health Research Centre, University of Almería, 4005 Almería, Spain; imf902@ual.es; 2Department of Sports Science, Nord University, 7600 Levanger, Norway

**Keywords:** ankle stiffness, contact time phase, foot angular velocity, sprint time, IMU devices

## Abstract

Foot stiffness is a modulator of sprint performance. However, studies that analysed foot angular velocities using inertial measuring units (IMU) for different events within the sprint contact time phase are scarce. The aim of this study was to investigate the relationship between angular foot step-by-step kinematics and sprint performance during a 50-metre sprint in experienced male and female sprinters. Foot kinematics were measured using IMU devices integrated with a 3-axis gyroscope and a laser gun. The main findings were that men performed faster sprints (6.11 ± 0.35 s vs. 6.77 ± 0.24 s), but the maximal angular foot kinematics were the same between sexes. Maximal angular velocities increased until strides 6–7, where they stabilized. Time from touchdown to maximal dorsiflexion velocity did not change between strides, whereas time from maximal dorsiflexion velocity to toe off decreased until stride 6. Plantarflexion velocities, especially in toe off, showed the greatest associations with sprint times, whereas maximal dorsiflexion velocity presented no association with sprint times. The time from dorsiflexion velocity to toe off from stride 7 onwards determined the sprint performance and was shorter for faster sprinters. The analysis of these variables provides essential information to athletes and coaches that may help to enhance the quality and efficiency of the sprint cycle by giving detailed information on each single stride of the sprint.

## 1. Introduction

Sprinting is an important ability that has been linked to performance for a range of sporting disciplines, such as football, rugby, and athletics [1,2], in all age groups [3]. Thus, it is an important training goal for strength and conditioning professionals to develop sprinting ability and technique in such athletes [1,2,4]. Methods to improve sprint performance are focused on improving athletes’ horizontal power output and force, and increasing technical efficiency [5], which requires drills such as “ankling” exercises [1,2,4].

During sprinting, it is observed that the work required to move the body in a forward direction is mostly modulated at first by the foot segment and thereafter by the hip [6], with the moment arm of the ground reaction force being the greatest in the ankle joint [7]. Since sprint performance is highly dependent on the maximal production of power from the first step of the sprint, it has been hypothesised that reducing dorsiflexion, with its concomitant increase in ankle plantar flexor moment, would enhance sprint time [5,7,8,9]. Leg stiffness could therefore be a modulator of sprint performance since a “stiffer” foot during the initial stance phase of the dorsiflexion, i.e., less dorsiflexion movement, would be a beneficial technical feature for horizontal velocity and therefore for the time of the sprint [5,6,9].

Earlier studies analysed the relationship between ankle plantarflexion stiffness and sprint performance using dynamometers [10,11,12], with the drawback being that joint stiffness data were not obtained from the actual sprint, which could lead to inconsistent outcomes. Furthermore, joint stiffness was assessed during the actual sprint using force platforms by combining a kinematic analysis with motion capture systems [5,7,13,14]. However, those methods are expensive, require a confined area and take a long time to process data. Thus, more accessible methods are needed to obtain such data that are not restricted to those limits.

Previous studies analysed the development of kinematics per step, assessing variables such as step velocity, flight time, step frequency, step length and contact times [2,15,16]. For example, van den Tillaar [2] investigated step and joint kinematics (maximal angular velocities at touchdown and toe off) on 30 m sprints using a laser gun and a contact mat but did not analyse the different stages within the contact time phase, which could provide more insights into foot and ankle stiffness, because those devices do not provide such information. This is because the contact time phase can be divided into three events: at touchdown the plantarflexion velocity, followed by a maximal dorsiflexion velocity (withstanding gravitational forces) and at toe off the maximal plantarflexion velocity again. Hence, the analysis of angular velocities at such events and the timings between them would help to provide some insight into foot and ankle performance and useful information for strength and conditioning professionals to enhance sprint performance [2].

Wearable inertial measurement units (IMUs) have previously been used to measure kinematic data, such as sprint times, angular velocities or sprint velocities [2,17,18]. For example, Arai, Obuchi, Shiba, Omuro, Nakano and Higashi [17] measured ankle joint kinematics with IMUs during jumping, and Struzik, Konieczny, Stawarz, Grzesik, Winiarski and Rokita [18] analysed the relationship between ankle kinematics and sprint times using IMUs. However, to the best of our knowledge, no study has assessed foot/ankle kinematics and angular velocities in different events within the contact phase of sprinting using IMUs [19,20]. The ease that IMU devices provide for the measurement of such kinematics could be an interesting tool for professionals when it comes to sprint performance. If IMUs can positively identify angular velocities at the contact phase for each stride in sprinting, and if this data can distinguish between fast and slower sprinters due to differences in foot and ankle performance, this could be an easy tool for enhancing training quality.

Therefore, the aim of the study was to investigate the relationship between angular foot/ankle step-by-step kinematics measured by IMUs and sprint performance during a 50 m sprint in experienced male and female sprinters. Furthermore, whether these foot/ankle step-by-step kinematics can distinguish between faster (shorter 50 m times) and slower sprinters was also studied in both sexes. It was hypothesised that faster plantarflexion velocities at touch down and toe off, and a lower angular velocity change to the maximal dorsiflexion velocity in between (incremental foot and ankle stiffness), would result in faster sprint times in men and women. If this is the case, this information at different phases of the total sprint distance could help trainers and athletes, via direct feedback after each sprint, to optimise their training and thereby enhance their sprint performance.

## 2. Materials and Methods

### 2.1. Study Design

To investigate the relationship between angular foot/ankle step-by-step kinematics measured by IMUs and sprint performance during a 50 m sprint in experienced male and female sprinters, a cross-sectional correlational study with a between-subject design was used, in which experienced male and female sprinters, ranging from national to international experience levels, were tested. Sex as well as fast and slow sprinters were the independent variables, while foot kinematics (angular velocity at touch down and toe off and peak dorsiflexion velocity during contact) and their timing were the dependent variables.

### 2.2. Participants

Seventeen experienced male sprinters (age 24.4 ± 7.8 years, body mass 77.1 ± 7.2 kg, body height 1.82 ± 0.07 m, training experience 13.5 ± 6.8, best 100m times of 11.15 ± 0.60 s) and eleven experienced female sprinters (age 20.4 ± 3.2 years, body mass 59.7 ± 5.1 kg, body height 1.68 ± 0.07 m, training experience 9.4 ± 3.4, best 100 m times of 12.54 ± 0.52 s), participated in this study. They were instructed to avoid undertaking any maximal sprint training in the 48 h prior to testing. Written consent was obtained from all participants, or parents when the participants were under the age of 18 years old, before participation. The study was conducted in accordance with the latest revision of the Declaration of Helsinki and current ethical regulations for research and was approved by the National Center for Research Data (pr.nr: 991974).

### 2.3. Procedure

After an individualised warm-up, each participant performed two 50 m sprints with a 6–10 min rest between each sprint on an indoor track to minimise the effect of weather conditions. The warm-up was individualised, since the subjects were experienced sprinters who knew what type of warm-up worked best for them to achieve their best sprinting performance and avoid possible injuries. Subjects initiated each sprint from a three-point start (one hand on the floor) in a split stance with the hand behind the line as this start is often used in regular training. Speed measurements and distances were continuously recorded during each attempt using a CMP3 Distance Sensor laser gun (Noptel Oy, Oulu, Finland), with sampling at 2.56 KHz. Total sprint times were derived from the laser distance over 50 m from the start.

Angular velocities and timing of the foot kinematics for each step throughout the sprint were derived from using wireless 9 degrees of freedom inertial measurement units (IMU) integrated with a 3-axis gyroscope (Ergotest Technology AS, Langesund, Norway) attached on top of the shoelaces of the spikes of each foot. The sampling rate of the gyroscope was 500 Hz, with a maximal measurement range of 2000°·s^−1^ ± 3%.

Angular foot kinematics were measured by the IMUs (gyroscopes) at three events: plantarflexion at first foot contact (touchdown), maximal dorsiflexion velocity and maximal plantarflexion velocity at toe off with the timing between the three events (Figure 1). Angular velocities measured with the gyroscopes on the top of the feet at touchdown and toe off were identified as highly accurate by previous studies [16,21].

Additionally, the differences between the angular velocities at touch down, maximal dorsiflexion velocity and at toe off were calculated. The pattern of maximal plantarflexion velocity followed by an abrupt dip (change to maximal dorsiflexion velocity) and the increase in plantarflexion to a maximum again were recognised as the start and end of the contact time of each step in an unpublished pilot study, which compared contact and flight time data measured with an infrared contact mat over 30 m (Ergotest Technology AS, Langesund, Norway) (ICC = 0.95) with force plates [21]. The laser gun (measuring the 30 m distance and time exactly) and IMUs were synchronised with the MuscleLab v10.202 (Ergotest Technology AS, Langesund, Norway). The step kinematics were evaluated for the first thirteen strides (one left and right step together) as all participants had at least that many strides to cover the 50 m distance.

### 2.4. Statistical Analyses

The normal data distribution was analysed using the Shapiro–Wilk normality test and the homoscedasticity of variances was tested using Levene’s test. Since all the variables presented a normal distribution and homogeneity of variance, parametric tests were performed. The mean values and standard deviations of the dependent variables were extracted from the descriptive statistics.

To investigate the correlations between the sprint times and ankle step kinematics, Pearson’s correlation was used for all participants and per sex. Threshold values for the correlation coefficients interpretation as an effect size were 0.1–0.3 (trivial), 0.3–0.5 (moderate), 0.5–0.7 (large), and 0.7–0.9 (very large) [22]. To compare the 50 m sprint times between sexes and angular kinematics over distance, a two-way ANOVA mixed design (2 sexes × 13 strides: repeated measures) was used. In addition, two-way ANOVA mixed designs (2 fast–slow sprinters) × 13 stride: repeated measures) were conducted for each of the sexes between the faster (shorter 50 m times) and slower sprinters of equal size (eight per group for men and five per group for women) on these parameters to investigate if foot kinematics can be distinguished at different strides and level. Post hoc comparisons with least mean difference were performed for pairwise comparison between each following stride of the sprints. Effect size was evaluated with partial eta squared (η_p_^2^), where 0.01 < η_p_^2^ < 0.06 constituted a small effect, 0.06 < η_p_^2^ < 0.14 a medium effect, and η_p_^2^ > 0.14 a large effect [23]. The level of significance was set at *p* < 0.05, and all data were expressed as mean ± standard deviation (SD). Analysis was performed with SPSS Statistics for Windows, version 27.0 (IBM Corp., Armonk, NY, USA).

## 3. Results

The sprint times were significantly shorter (*p* < 0.001) in men (6.11 ± 0.35 s) in comparison to women (6.77 ± 0.24 s).

No significant differences for any of the kinematic parameters were found between sexes (F ≤ 1.6, *p* ≥ 0.169, η_p_^2^ ≤ 0.09), while maximal angular dorsiflexion and plantarflexion velocity at touchdown and toe off increased significantly with every stride (F ≥ 11.7, *p* ≤ 0.001, η_p_^2^ ≤ 0.41) until stride 6–7, where they stabilised (Figure 2). The time from touchdown to maximal dorsiflexion velocity did not change significantly over the strides (F = 2.09, *p* = 0.092, η_p_^2^ = 0.11), while the times from maximal dorsiflexion velocity to toe off significantly decreased with each stride for the first six strides (F = 115, *p* < 0.001, η_p_^2^ = 0.87). No significant sex–stride interaction effect was found for any of the variables (F ≤ 2.07, *p* ≥ 0.117, η_p_^2^ ≤ 0.11, Figure 2).

Pearson’s correlation coefficients showed moderate to large associations between sprint time and ankle kinematics when all strides were taken together in the whole group and specified for men and women for most variables, except for maximal dorsiflexion velocity, and time from touchdown to maximal dorsiflexion velocity (Table 1).

When analysing relationships between sprint time and ankle kinematics per stride, toe off velocity moderately correlated with almost each step, while the largest correlation between touchdown velocity and sprint times was observed at the first stride in both sexes (Table 1). These higher plantar velocities at touchdown and toe off also resulted in moderate to large correlations in velocity changes to and from the maximal dorsiflexion velocity, especially from stride 8 to 13. This also resulted in significant moderate to large positive correlations for stride 7 onwards in both sexes in the time from maximal dorsiflexion velocity to toe off and the total contact time (touchdown–toe off) (Table 1).

When comparing the eight faster sprinters to the eight slower sprinters in the male group and the five faster sprinters to the five slower sprinters in the female group, some kinematic differences were noted. Evidently, sprint times significantly differed (*p* < 0.001) between fast and slow sprinters in both men (5.82 ± 0.13 s vs. 6.39 ± 0.28 s) and women (6.57 ± 0.18 s vs. 6.97 ± 0.10 s). Furthermore, the faster male sprinters presented higher touchdown plantarflexion velocity during stride 1 (F = 7.45, *p* = 0.011, η_p_^2^ = 0.199) and higher toe off velocities during all strides (F ≥ 5.54, *p* ≤ 0.025, η_p_^2^ ≥ 0.156) compared to the slower male sprinters (Figure 3). Times between maximal dorsiflexion velocity to toe off during strides 8 to 13 (F ≥ 7.00, *p* ≤ 0.013, η_p_^2^ ≥ 0.189) (Figure 3), and total contact time from touchdown to toe off during strides 8 to 13 (F ≥ 6.86, *p* ≤ 0.014, η_p_^2^ ≥ 0.186), were significantly shorter for the faster men in comparison to the slower men.

In contrast, women presented no significant differences in maximal velocities between the faster and slower sprinters (*p* > 0.05) (Figure 4). However, times from maximal dorsiflexion velocity to toe off during strides 11 to 13 (F ≥ 5.40, *p* ≤ 0.034, η_p_^2^ ≥ 0.252) were significantly shorter for the faster women in comparison to slower women (Figure 4). No significant level–stride interaction effects were found for any of the variables for each sex (F ≤ 1.48, *p* ≥ 0.114, η_p_^2^ ≤ 0.15).

## 4. Discussion

The purpose of the current study was to investigate the relationship of angular foot step-by-step kinematics and sprint performance during a 50 m sprint in experienced sprinters. The main findings were that men performed faster sprints than women, but the maximal angular foot kinematics were similar between sexes. Maximal angular velocities increased until strides 6–7, where they stabilised. Moreover, the time from touchdown to maximal dorsiflexion velocity was constant over strides, whereas the time from maximal dorsiflexion velocity to toe off decreased until stride 6. Plantarflexion velocities showed the greatest associations with sprint times, especially at toe off, whereas maximal dorsiflexion velocity showed no association with sprint times. Time from maximal dorsiflexion velocity to toe off from stride 7 onwards determined sprint performance and differentiated between faster and slower sprinters.

The absence of kinematic differences between sexes can be explained by the geometric scaling paradigm [24,25]. Angular velocities presented no differences between sexes, and as men are generally taller than women, it is expected that men have longer feet and legs. Consequently, they cause higher linear joint movements and faster movements to the ground [25]. Therefore, men developed faster linear velocity for the same angular velocity than their female counterparts due to their longer levers [24,26].

A significant negative association between maximal plantarflexion velocity at toe off and sprint times was shown that followed our hypothesis, which was also visible in the comparison between faster and slower men. Higher toe off velocities are explained by the greater ankle plantar flexor impulse that faster sprinters display, as reported previously [6,9,11]. The higher plantarflexion velocity at toe off gives faster athletes the ability to create more propulsive impulse which results in faster sprints [27].

The plantarflexion velocity at touchdown showed the highest correlation to sprint times at the first stride, which is in accordance with previous studies showing that better sprint performance was associated with greater propulsive impulse at the initial acceleration phase of the sprint [3]. This higher velocity at the first stride enables good early acceleration due to the exertion of the maximal force from the beginning of the sprint [28]. The first two steps (first stride) velocity determines the performance of a successful sprint start and initial acceleration phase, with longer pushing and contact times but shorter flight times, with the aim of increasing the velocity as quickly as possible [28,29]. Therefore, a successful sprint start, with its concomitant peak angular plantarflexion at touchdown and toe off in the initial strides, would facilitate a shorter sprint time and higher acceleration [30]. This is seemingly related to a higher frequency of strides during the sprint, promoting shorter contact times in the middle and last part of the sprint, once the maximal sprint velocity is reached [30,31].

Another key finding of the present study was the absence of significant moderate/strong correlations for maximal dorsiflexion velocity and time from touchdown to maximal dorsiflexion velocity (braking phase) with sprint times. Despite the faster plantarflexion shown in faster sprinters, this unexpectedly showed no connection to a reduced dorsiflexion velocity, so the previous theory relating stiffer ankles to faster sprinters did not explain the best sprint times. Such parameters are similar between faster and slower sprinters, and foot stiffness during this phase remains constant throughout the strides for all sprinters based on angular velocities and times. Previous studies that associated leg stiffness with better sprint times might have overlooked the influence of underlying negative work on all muscles during this phase, since most studies did not measure foot stiffness by assessing joint movements during the actual sprint [3,10,11,12].

During the negative work phase (from touchdown to maximal dorsiflexion), muscles need to resist an overload of the body mass driving in a forward direction plus the gravity component. As for any stretch-shortening cycle (SSC), the spring-like motion of the ankle is also modulated by the range of motion of the lever arm of the involved joints, so the sprinter would need to overtake such a load within a specific range of motion [30,31]. The similar time and velocity change from touchdown to maximal dorsiflexion velocity presented in all sprinters could perhaps be explained by the neural control of the SSC [32]. The SSC performance is directly associated with the energy storage of the elastic elements of the muscle, which is modulated by the pre-activated muscle–tendon complex. Therefore, it is likely that better sprinters elicit optimised pro-active and reactive motor strategies, simultaneously anticipating the motor response to the touchdown moment. Eventually, due to the anticipation mechanism, better sprinters seem to be capable of exerting a greater linear impulse despite having a similar braking phase in terms of time and change in velocity [32,33].

From approximately stride 7 of the sprint and onward, times from the maximal dorsiflexion velocity to the maximal plantarflexion velocity at toe off stabilise because maximal sprint velocity was also reached around these strides [2] and more strongly determined sprint performance. The significant correlations to sprint time draw a distinction between the faster and slower sprinters. The capacity of the athlete to maintain a maximal velocity over time is crucial for sprint performance [6,34]. The greater correlations shown by faster sprinters from stride 7 onwards highlight the importance of being able to maintain a maximal velocity during the non-acceleration phase of the sprint. Sprinters’ capability of reaching the maximal velocity as quickly as possible and maintaining it throughout a sprint achieved the best sprint times [4,19,20]. Hence, by performing this method, professionals and athletes could accurately determine the strides where their sprint performance was not the best possible (lower peak angular velocity at touch down or toe off compared to other strides or higher-level athletes). Thereby the athletes could potentially optimise their training drills by using this information immediately after each sprint.

Higher step frequency, shorter contact time and increases in flight time were previously established as determinants of running velocity [15,34,35]. The findings of the current study add valuable information to the body of knowledge on this topic, which has already confirmed the foot and ankle as key sources with which to maximise sprint performance by means of rapidly accelerating in a forward direction [6,8,34,35]. Although the hypothesis about incremental foot stiffness (higher plantarflexion velocities at touch down and toe off and a lower angular velocity change to maximal dorsiflexion velocity in between) was incorrect, faster toe off velocities and shorter times from maximal dorsiflexion velocity to toe off are still related to sprint performance. This matches the theory that a larger plantar flexor torque, especially at maximal velocity, is credited to a stiffer spring-like foot movement from midstance to the toe off phase [8,9]. Then, understanding this method as being reliable and useful in practice, professionals should be able to easily measure such variables to identify decisive points of the contact phase of the sprint and manage the training according to the demands of each individual athlete [18,20].

The current study has some limitations. Since we only had IMUs on top of the spikes and not on the leg, no exact angular ankle joint velocity was measured, only the foot rotation. However, the gyroscope measurements are still an indication of the ankle kinematics as the foot rotates as part of the ankle joint. In addition, the sample size was relatively small, and the sprint performances of the sprinters are not at an international level, especially for the group of women, which limits the generalisation of the findings to all sprint levels. A larger sample size, including different levels of athletes, would likely consolidate the results obtained for both sexes. Additionally, the cross-sectional design prevents direct causal relationships from being established. As a consequence, future studies should involve more women to obtain a balanced sample size on gender and higher-level athletes. It should also include a training period with a focus on increasing maximal plantarflexion velocity during sprints to investigate whether this could enhance sprint performance directly.

## 5. Conclusions

To the best of our knowledge, this is the first study to analyse angular foot kinematic parameters during the three different events of the contact phase (touchdown, maximal dorsiflexion velocity and toe off) with IMUs. In summary, it is concluded that foot kinematics during the contact phase of a sprint for each stride provide useful information on the foot performance during the actual sprint. The analysis of such variables reports essential information that may help to enhance the quality and efficiency of the sprint cycle by giving detailed information on each single stride of the sprint. Furthermore, emerging methods that could make it easier to analyse and interpret such variables, by developing an algorithm that can automatically detect these parameters, might be elaborated in further research. Therefore, by direct feedback after each sprint, this could improve the quality of training and likely enhance performance more quickly over a period of time. Eventually, professionals in this field will be able to optimize and individualize training drills in real-time to maximize sprint performance.

## Figures and Tables

**Figure 1 ijerph-19-06786-f001:**
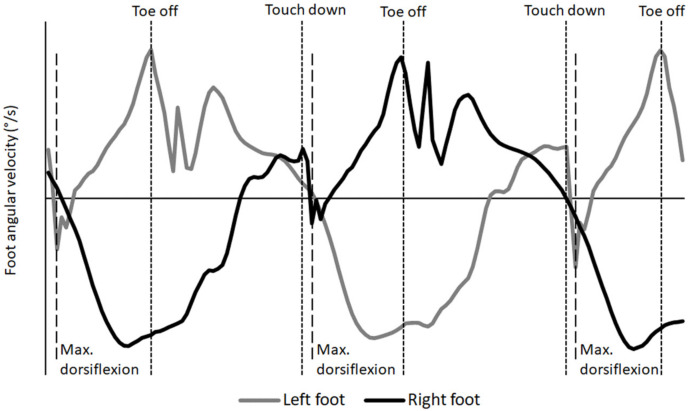
Illustration of the angular foot kinematics at the three different events during the contact time phase in three steps of the sprint.

**Figure 2 ijerph-19-06786-f002:**
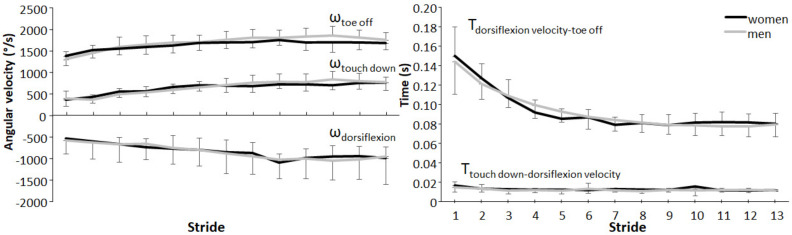
Comparison of the angular velocities (±SD) and time variables for each stride averaged for men and women.

**Figure 3 ijerph-19-06786-f003:**
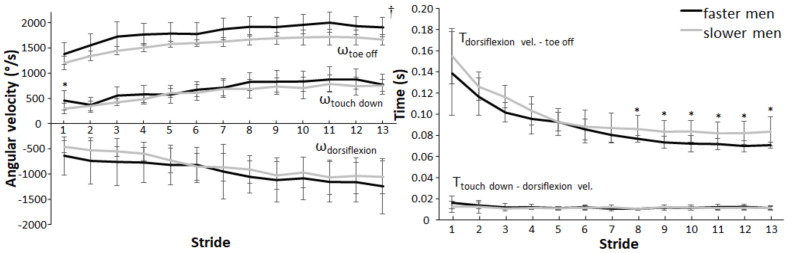
Comparison of the angular velocities (±SD) (**left**) and time variables (±SD) (**right**) for each stride between faster and slower men. * Indicates a significant difference between faster and slower men for this stride (*p* < 0.05). † Indicates a significant difference between faster and slower men for each stride (*p* < 0.05).

**Figure 4 ijerph-19-06786-f004:**
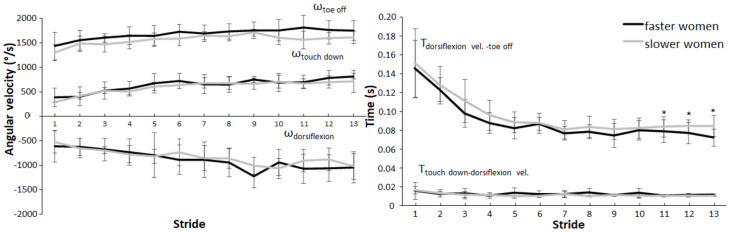
Comparison of the angular velocities (±SD) (**left**) and time variables (±SD) (**right**) for all individual strides between faster and slower women. * Indicates a significant difference between fast and slow women for this stride (*p* < 0.05).

**Table 1 ijerph-19-06786-t001:** Pearson’s bivariate correlations analysis evaluating associations between velocities and timing variables with sprint times per stride.

		Correlations with 50 m Sprint Times													
	Stride	1	2	3	4	5	6	7	8	9	10	11	12	13	All
**Men**	Touchdown velocity	−0.65 *	−0.36	−0.57 *	−0.61 *	−0.19	−0.51 *	−0.37	−0.48 *	−0.47	−0.40	−0.38	−0.43	−0.18	−0.51 *
	Maximal dorsiflexion velocity	0.13	0.18	−0.01	0.03	−0.03	−0.15	−0.13	0.00	−0.04	−0.10	−0.01	0.05	−0.18	−0.06
	Toe off velocity	−0.47	−0.48	−0.54 *	−0.59 *	−0.48 *	−0.51 *	−0.59 *	−0.61 *	−0.62 *	−0.64 *	−0.66 *	−0.52 *	−0.53	−0.63 *
	Velocity change from touchdown to dip	−0.48	−0.48	−0.33	−0.53 *	−0.26	−0.28	−0.10	−0.41	−0.38	−0.32	−0.34	−0.38	−0.39	−0.40
	Velocity change from dip to toe off	−0.42	−0.51 *	−0.37	−0.55 *	−0.37	−0.30	−0.19	−0.46	−0.43	−0.40	−0.44	−0.39	−0.56 *	−0.47
	Time from touchdown to dip	−0.43	−0.22	−0.31	−0.17	−0.25	−0.01	0.33	−0.08	−0.06	−0.05	−0.18	−0.34	−0.27	−0.28
	Time from dip to toe off	0.27	0.27	0.43	0.35	0.27	0.25	0.42	0.56 *	0.64 *	0.63 *	0.59 *	0.70 *	0.47	0.46
	Time from touchdown to toe off	0.19	0.21	0.38	0.32	0.24	0.26	0.46	0.55 *	0.64 *	0.64 *	0.56 *	0.62 *	0.42	0.42
**Women**	Touchdown velocity	−0.65 *	0.00	−0.20	−0.39	−0.48	−0.66 *	−0.08	0.06	−0.53	0.00	−0.20	−0.47	−0.14	−0.28
	Maximal dorsiflexion velocity	0.37	0.07	0.26	0.12	0.24	0.57	0.29	0.43	0.69 *	0.00	0.36	0.44	0.09	0.35
	Toe off velocity	−0.40	−0.31	−0.54	−0.40	−0.43	−0.63 *	−0.51	−0.65 *	−0.27	−0.56	−0.73 *	−0.57	−0.63 *	−0.67 *
	Velocity change from touchdown to dip	−0.52	−0.08	−0.31	−0.30	−0.36	−0.67 *	−0.26	−0.33	−0.74 *	0.00	−0.38	−0.57	−0.18	−0.40
	Velocity change from dip to toe off	−0.46	−0.21	−0.56	−0.34	−0.35	−0.63 *	−0.42	−0.60 *	−0.71 *	−0.32	−0.69 *	−0.60	−0.35	−0.53
	Time from touchdown to dip	0.23	0.34	0.25	0.35	−0.41	0.15	0.30	−0.41	0.20	−0.32	−0.10	0.05	−0.42	0.04
	Time from dip to toe off	0.40	0.32	0.43	0.52	0.46	0.32	0.44	0.60	0.50	0.50	0.48	0.63	0.80 *	0.57
	Time from touchdown to toe off	0.47	0.38	0.49	0.55	0.28	0.34	0.53	0.42	0.51	0.04	0.47	0.58	0.77 *	0.52
**Whole Group**	Touchdown velocity	−0.53 *	0.00	−0.16	−0.33	−0.03	−0.25	−0.23	−0.40 *	−0.45 *	−0.31	−0.48 *	−0.40 *	−0.14	−0.40 *
	Maximal dorsiflexion velocity	0.20	0.14	0.02	−0.06	0.01	0.03	0.00	0.14	−0.02	−0.05	0.14	0.16	−0.12	0.05
	Toe off velocity	−0.15	−0.15	−0.44 *	−0.48 *	−0.46 *	−0.40 *	−0.51 *	−0.61 *	−0.50 *	−0.64 *	−0.68 *	−0.57 *	−0.54 *	−0.59 *
	Velocity change from touchdown to dip	−0.42 *	−0.29	−0.20	−0.26	−0.15	−0.28	−0.17	−0.47 *	−0.36	−0.31	−0.47 *	−0.43 *	−0.37	−0.41 *
	Velocity change from dip to toe off	−0.26	−0.29	−0.34	−0.37	−0.33	−0.34	−0.26	−0.54 *	−0.38	−0.44 *	−0.57 *	−0.49 *	−0.53 *	−0.49 *
	Time from touchdown to dip	0.00	−0.07	0.07	0.06	−0.06	−0.10	0.39	0.09	−0.08	0.07	−0.26	−0.31	−0.17	0.06
	Time from dip to toe off	0.28	0.33	0.23	0.07	−0.03	0.18	0.12	0.38	0.43 *	0.52 *	0.54 *	0.63 *	0.44 *	0.39 *
	Time from touchdown to toe off	0.28	0.31	0.25	0.08	−0.04	0.14	0.20	0.40 *	0.42 *	0.45 *	0.50 *	0.55 *	0.41 *	0.38 *

* *p* < 0.05; dip = maximal dorsiflexion velocity.

## Data Availability

The raw data supporting the conclusions of this article will be made available by the authors without undue reservation.

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
