# Peer review of "Relationship between Step-by-Step Foot Kinematics and Sprint Performance"

_ijerph, 2022, doi:10.3390/ijerph19116786_

Round 1

Reviewer 1 Report

We thank the authors for taking efforts to revise the manuscript. I understand that I have a strong point of accusation. However, please take note that I am delivering the discussion and comments as a matter of fact on the content and what is presented in the manuscript. I believe that the truth of the science and thus improvement is based on the matter of fact and not based on and shall not be aware of whether the authors are experts and authorities of the field. In fact, the analysis could be adaptative to the results found but the overall strategy itself shall be preplanned and presented well to avoid potential bias (i.e., “tweaking” the analysis based on what got). The authors said that because they found that there is no significance or not important and therefore they do not include them or weakly presented them in the manuscript.  The backend information is only known by the authors but not the readers. This is not the blame on the reviewer to challenge the analysis method if the authors are hiding out their full analysis strategy in the first place while squeezing it piece by piece when queries are coming out.

For the faster/slower sprinter issues, I believed the authors meant that they divided the individuals into faster and slower groups of “equal size” for each sex (therefore, the point is to equally divided, not based on a cut-off of velocity). Please make a standalone statement on the grouping issues before putting forward the statistics for clarity. I agree on the use of LSD.

I agree the analysis of the faster/slow had added value to the manuscript. However, please reinforce the objective or additional scope at the end of the introduction to avoid an impression of data dredging and data fishing.

I yet disagree with the fact that the authors applied statistics to make a distinction on a variable. While the authors mention that they are statisticians, I do think that the authors agree that ANOVA is primarily used as inferential statistics (i.e. to find if there is any significant difference/association between sth, and infer to the population) and it deems not appropriate and comfortable to say that ANOVA is used for other purposes that give confusion, at least formally in the Methods – Statistical Analysis section. However, I agree that the authors may base on the meaning of the statistics and make a further interpretation of the “distinguishment” in the discussion.

Author Response

Dear Reviewer,

Thank you for reviewing the manuscript again. We have changed the manuscript according to the last comments of the reviewer and think that it is now suitable for publication.

Reviewer 1

We thank the authors for taking efforts to revise the manuscript. I understand that I have a strong point of accusation. However, please take note that I am delivering the discussion and comments as a matter of fact on the content and what is presented in the manuscript. I believe that the truth of the science and thus improvement is based on the matter of fact and not based on and shall not be aware of whether the authors are experts and authorities of the field. In fact, the analysis could be adaptative to the results found but the overall strategy itself shall be preplanned and presented well to avoid potential bias (i.e., “tweaking” the analysis based on what got). The authors said that because they found that there is no significance or not important and therefore they do not include them or weakly presented them in the manuscript.  The backend information is only known by the authors but not the readers. This is not the blame on the reviewer to challenge the analysis method if the authors are hiding out their full analysis strategy in the first place while squeezing it piece by piece when queries are coming out.

For the faster/slower sprinter issues, I believed the authors meant that they divided the individuals into faster and slower groups of “equal size” for each sex (therefore, the point is to equally divided, not based on a cut-off of velocity). Please make a standalone statement on the grouping issues before putting forward the statistics for clarity. I agree on the use of LSD.

We have included equal size to the text to avoid confusion. We have added an extra sentence at the end of the introduction about distinguishes between faster and slower sprinter.

I agree the analysis of the faster/slow had added value to the manuscript. However, please reinforce the objective or additional scope at the end of the introduction to avoid an impression of data dredging and data fishing.

We have added an additional scope to the end of the introduction to avoid fishing. We hope this increases the purposes of our analyses.

I yet disagree with the fact that the authors applied statistics to make a distinction on a variable. While the authors mention that they are statisticians, I do think that the authors agree that ANOVA is primarily used as inferential statistics (i.e. to find if there is any significant difference/association between sth, and infer to the population) and it deems not appropriate and comfortable to say that ANOVA is used for other purposes that give confusion, at least formally in the Methods – Statistical Analysis section. However, I agree that the authors may base on the meaning of the statistics and make a further interpretation of the “distinguishment” in the discussion.

We agree that ANOVAs is primarily used as inferential statistics. That is also why we only used it to investigate between sexes and if the kinematics changes over the 50m. As said before it was not necessary to perform an ANOVA between the faster and slower sprinters, but we think it helps the coaches to understand the relationships found easier. Since we have added the part of distinguish between the faster and slower sprinters in the introduction it is easier to agree that we performed also ANOVAs.

Reviewer 2 Report

The authors modified the text appropriately in response to the requests I had made. In my opinion, the article is now much more understandable.

Author Response

Thank you for reviewing the manuscript

This manuscript is a resubmission of an earlier submission. The following is a list of the peer review reports and author responses from that submission.

Round 1

Reviewer 1 Report

The authors investigated the relationship between kinematics and (gender or level of sprinters??) in sprint performance. I believed that the statistical methods were not correct, while the description of the study design and dependent/independent factors were very confusing (though the authors declared them in the methods). I am not confident that the conclusions were valid. In precise, I am not sure whether the level of sprinters (i.e., faster/slower sprinter named in the text) were dependent or independent factors based on the description of Methods. If this was the independent factor, whether this factor is stratified with gender or nested with gender is not understood. There was no information on the nested groups or the subgroup. Moreover, I did not understand how the authors defined sprint distance, while I believed that should be stride length for each stride.

From the results, there was another factor (longitudinal time factor). I am not sure whether the authors referred to the findings as six strides or the sixth stride, which gave a totally different meaning in the analysis. In this case, it seemed to be a 3-way ANOVA with (gender x level x time), but it was not.

In Line 145, the authors mentioned that they would compare time with sexes on kinematics using two-way ANOVA, which was incorrect. In such case, a two-way ANOVA mixed design shall be used. Therefore, I did not think that the results are valid given the wrong statistical method. In addition, I do not view the two-way ANOVA as correct without considering interaction, main effects, and simple main effects. In Line 146, the authors compared the faster and slower sprinters for each sex. Therefore, there were two independent factors. However, the authors used the one-way ANOVA, which did not seem to be correct.

For all the ANOVAs and multiple tests for different outcome parameters, all pairwise comparisons and multiple parameter tests shall be accompanied by an adjustment method clearly in the Methods and Results section. I am not sure if the authors indeed took these into consideration; otherwise, the statistical methods were not correct.

Some other specific comments are listed below:

Line 8: It is too aggressive to mention that “no studies”  had done something, unless you can prove it. It will be okay if there is a recent review mentioned that they cant find those studies.

Line 37: Please link leg stiffness to joint moment.

Line 45: “with video analysis with motion analysis” could not be understood. Nevertheless, the authors mixed kinematics and kinetics. Please make sure they are correctly named.

Line 52: How can one use a laser gun to investigate joint kinematics?

Line 68: For “contact time phase”, please use a correct formal term either in the field of gait analysis or sprint.

Line 77: Please defined the outcome measures in the last paragraphs and give abbreviations to avoid the clumsy sentences here. In addition, please justify why these outcome measures were important to answer your research question. Why is it important to study step-by-step kinematics?

Line 81: “via the autodetection of this kinematics” is redundant.

 Line 95: There was no information on the level of athletes (faster/slower sprinter)

Line 113: What are the specifications of the IMU? Did you purchase a system? Or work on the sensors in-house. Either way, the brand and series of the sensor/system shall be addressed.

Line 129: Distance between what and what?

Line 135: Please address the role of the laser gun.

Line 162: No need to report effect size if the results were not significant

Line 211: “were the same” is too aggressive. I did not think that the results could be exactly the same for a continuous variable.

Line 273: Significant correlation did not mean that the variable had strong distinctive power. That need to be used by another statistical method.  

Line 289: The research hypothesis shall address the outcome parameters of the research. There is no “foot stiffness” parameters in the research. Please revise it and use extrapolation towards the implicated endpoint.

Line 303: shall be different levels of athletes

Line 304: The study design was not clearly stated leading to confusion here. The authors utilized a longitudinal design (strides). It is not very clear that the statement here mentioned “cross-sectional designs”.

 Line 306: Not understand “more women shall be involved”. I think the authors mean a balanced sample size on gender.

Line 326: Despite that raw data may not be obtained directly. I do recommend the authors upload the mean(SD) vs time data points information for Figure 2 to Figure 4 in supplementary, such that other researchers could have numeric data to review your article or do a meta.

Author Response

We want to thank the reviewers for their comments to the manuscript. We have answered to all comments of the reviewers and think that the manuscript now it suitable for publication. All changes in the text are colored red.

Reviewer 1

The authors investigated the relationship between kinematics and (gender or level of sprinters??) in sprint performance. I believed that the statistical methods were not correct, while the description of the study design and dependent/independent factors were very confusing (though the authors declared them in the methods). I am not confident that the conclusions were valid. In precise, I am not sure whether the level of sprinters (i.e., faster/slower sprinter named in the text) were dependent or independent factors based on the description of Methods. If this was the independent factor, whether this factor is stratified with gender or nested with gender is not understood. There was no information on the nested groups or the subgroup. Moreover, I did not understand how the authors defined sprint distance, while I believed that should be stride length for each stride.

We understand the confusion. We wanted to investigate the relationship between kinematics and gender in sprint performance. So, we performed correlations to investigate relationship between the kinematic variables and sprint time. Since we did not know if the kinematics were the same between men and women a 2-way ANOVA with repeated measures on strides (13) was performed to investigate if men and women had similar kinematics over the different strides. This is now changed in the statistical analysis to avoid confusion. Since no differences in kinematics between men and women were found all data could be used for the Pearson correlations. These correlations with times showed that some kinematics had a significant correlation with times indicating that when sprinting faster or slower this kinematic variable also changes. Since we think that just performing correlations between time and kinematics don’t easy show that when sprinting faster or slower is shown by changes in the foot kinematics. Therefore, we also analyzed the fastest with the slowest sprinters in each sexe to indicate what variables change and at which stride. We think that this gives more information than just performing correlations. Therefore, we have changed the order of the results. First kinematics between men and women, then correlations and in the end comparison of fast and slow sprinters to investigate more closely the evt. Differences in kinematics between faster and slower sprinters.

From the results, there was another factor (longitudinal time factor). I am not sure whether the authors referred to the findings as six strides or the sixth stride, which gave a totally different meaning in the analysis. In this case, it seemed to be a 3-way ANOVA with (gender x level x time), but it was not.

We have mentioned this now in the previous comment. We have performed a 2 (sexe) x 13 (strides, repeated measures) ANOVA. This is mentioned in the text now. We have also reported the interaction effects which were not significant for any of the variables. We did not want to perform a 3-way ANOVA, because then parts of the data will be gone (2 men and 1 woman in the middle). Since we have not so many subjects, we think that this is not so smart to do. As mentioned before the main reason of the study was to investigate correlations between kinematics and times. The analysis between the fast and the “slow” sprinters is an extra analysis to show in which strides differences clearly occur. We think this gives more information than just correlations. We have changed the order in the results to show that correlations are a start of the analysis, followed by the evt. Differences between fast and slow sprinters. We hope the reviewer is satisfied with these changes. Differences between fast and slow sprinters is a 2way ANOVA (mixed model). We have changed this also in the text.

In Line 145, the authors mentioned that they would compare time with sexes on kinematics using two-way ANOVA, which was incorrect. In such case, a two-way ANOVA mixed design shall be used. Therefore, I did not think that the results are valid given the wrong statistical method. In addition, I do not view the two-way ANOVA as correct without considering interaction, main effects, and simple main effects. In Line 146, the authors compared the faster and slower sprinters for each sex. Therefore, there were two independent factors. However, the authors used the one-way ANOVA, which did not seem to be correct.

Sorry for the confusion. The reviewer is correct that we need to use an ANOVA mixed design. We had used it but did not write it very clear. This is now changed.

For all the ANOVAs and multiple tests for different outcome parameters, all pairwise comparisons and multiple parameter tests shall be accompanied by an adjustment method clearly in the Methods and Results section. I am not sure if the authors indeed took these into consideration; otherwise, the statistical methods were not correct.

We fully agree with the reviewer that when performing post hoc analysis (pairwise comparisons) with many strides. We have now included what kind of post hoc analysis we did to the text.

Some other specific comments are listed below:

Line 8: It is too aggressive to mention that “no studies”  had done something, unless you can prove it. It will be okay if there is a recent review mentioned that they cant find those studies.

We have changed it in that studies analysing this are scarce.

Line 37: Please link leg stiffness to joint moment.

We have included “i.e. less dorsiflexion movement would..” to the text to link it to joint movements

Line 45: “with video analysis with motion analysis” could not be understood. Nevertheless, the authors mixed kinematics and kinetics. Please make sure they are correctly named.

We have changed it in “… with kinematic analysis with motion capture systems”

Line 52: How can one use a laser gun to investigate joint kinematics?

We agree you can’t measure joint kinematics with a laser, but it is mentioned STEP and joint kinematics. And in step kinematics it is step length that is measured with the laser.

Line 68: For “contact time phase”, please use a correct formal term either in the field of gait analysis or sprint.

We have changed it in contact phase

Line 77: Please defined the outcome measures in the last paragraphs and give abbreviations to avoid the clumsy sentences here. In addition, please justify why these outcome measures were important to answer your research question. Why is it important to study step-by-step kinematics?

We have rewritten the part on angular velocities in: faster plantarflexion velocities at touch down and toe off and a lower angular velocity change to maximal dorsiflexion velocity in between (incremental foot and ankle stiffness) … Furthermore, we have added … at different phases of the total sprint distance implying why step-by step kinematics is important.

Line 81: “via the autodetection of this kinematics” is redundant.

We have deleted this from the text now.

 Line 95: There was no information on the level of athletes (faster/slower sprinter)

We did not mention it here but only in the results part, where the different sprint times were given of the fast and slow sprinters. The anthropometric data was not different between the groups and therefore not specified either to avoid to much information, which is not important for answering the main question.

Line 113: What are the specifications of the IMU? Did you purchase a system? Or work on the sensors in-house. Either way, the brand and series of the sensor/system shall be addressed.

This was mentioned at the end of the sentence (Ergotest Technology AS, Langesund, Norway). We have moved it to earlier in the sentence for better information.

Line 129: Distance between what and what?

We have changed it in: differences in the angular velocities at touch down, maximal dirsoflexion velocity and at toe off to avoid confusion.

Line 135: Please address the role of the laser gun.

We have included (measuring the 30m distance and time exactly) to the text to explain the role of the laser.

Line 162: No need to report effect size if the results were not significant.

We agree that when results are not significant effect sizes most of the time are low. However, due to a limited n it is still possible that with non-significant differences effect-sizes can be medium. Also, many readers ask for what the effect size is. That is why we think it is also important to mention the effects sizes when it is not significant. We hope the reviewer understands our point of view.

Line 211: “were the same” is too aggressive. I did not think that the results could be exactly the same for a continuous variable.

We have changed it in similar to avoid of being too aggressive.

Line 273: Significant correlation did not mean that the variable had strong distinctive power. That need to be used by another statistical method.  

That is also why we performed an analysis on the fast and slower sprinters to show if the correlation was clear when analysing the faster with the slower sprinter.

Line 289: The research hypothesis shall address the outcome parameters of the research. There is no “foot stiffness” parameters in the research. Please revise it and use extrapolation towards the implicated endpoint.

We have specified, what we mean with “foot stiffness” in the introduction (higher plantarflexion velocities at touch down and toe off and a lower angular velocity change to maximal dorsiflexion velocity in between). We have now also written this in the discussion to avoid confusion.

Line 303: shall be different levels of athletes

Changed in different levels of athletes

Line 304: The study design was not clearly stated leading to confusion here. The authors utilized a longitudinal design (strides). It is not very clear that the statement here mentioned “cross-sectional designs”.

We mean with this that we only measured in a cross-sectional design in which we only measure at one moment in time. A longitudinal study in which we would measure at different times during a year or before and after a training period, would give us information about if we could change this variable correctly to enhance sprint performance. Thereby see if it the relationship is causal, when changing some of the parameters. We did not have a longitudinal design, only a design in which different strides were compared with each other (repeated measures), but these were not measured at different times to see the effect of f.e. training.

 Line 306: Not understand “more women shall be involved”. I think the authors mean a balanced sample size on gender.

We have included balanced sample size to the text now.

Line 326: Despite that raw data may not be obtained directly. I do recommend the authors upload the mean(SD) vs time data points information for Figure 2 to Figure 4 in supplementary, such that other researchers could have numeric data to review your article or do a meta.

We have uploaded mean and SD of figure 2-4 for other researchers.

Reviewer 2 Report

Basic reporting

The study assessed the relationship between angular foot step-by-step kinematics and sprint performance during a 50-metre sprint in experienced sprinters. The study is well written and is interesting from a practical point of view. I would recommend the authors address some amendments to improve the scientific quality of the paper.

ABSTRACT

In my opinion, should be added a final sentence highlighting the practical implications of the study.

INTRODUCTION

It is a well-written introduction supported with appropriate references. However greater emphasis and focus specifically on track and field athletes would strengthen the rationale for the study. 

Double check the Journal’s references style. I’m referring to the names of authors in lines 64 and 65.

MATERIAL AND METHODS

Participants:

L95-97Add the Training experience (years)

Did you check the sample size? The number of participants is pretty different from the male and female groups.

DISCUSSION

I think authors should compare more their findings with current literature. Especially with studies that have analysed similar tests in track and field athletes and also, with team-sports athletes.

I suggest you include more considerations about the limitation of your study.  Please add also which are the possible future lines.

CONCLUSIONS

The conclusion section should be further developed.

Moreover, a practical implications section should be included. How these findings can help track and field coaches and practitioners? Which practical implications can it have in training sessions? 

Author Response

We want to thank the reviewers for their comments to the manuscript. We have answered to all comments of the reviewers and think that the manuscript now it suitable for publication. All changes in the text are colored red.

Basic reporting

The study assessed the relationship between angular foot step-by-step kinematics and sprint performance during a 50-metre sprint in experienced sprinters. The study is well written and is interesting from a practical point of view. I would recommend the authors address some amendments to improve the scientific quality of the paper.

Thank you for this.

ABSTRACT

In my opinion, should be added a final sentence highlighting the practical implications of the study.

 We have added this sentence to the text: The analysis of these variables reports essential information to athletes and coaches that may help to enhance the quality and efficiency of the sprint cycle by giving detailed information of each single stride of the sprint.

INTRODUCTION

It is a well-written introduction supported with appropriate references. However greater emphasis and focus specifically on track and field athletes would strengthen the rationale for the study. 

Double check the Journal’s references style. I’m referring to the names of authors in lines 64 and 65.

We have checked the references style, and it does not state how many authors there should be mentioned in a citation.

MATERIAL AND METHODS

Participants:

L95-97Add the Training experience (years)

Included now.

Did you check the sample size? The number of participants is pretty different from the male and female groups.

Unfortunately, we did not, which is a main limitation of the study. This is mentioned in the text as a limitation. We wanted to have women from a certain level and in this region there were not more women available at this level. Sorry.

 DISCUSSION

I think authors should compare more their findings with current literature. Especially with studies that have analysed similar tests in track and field athletes and also, with team-sports athletes.

We wanted to check it with other studies. However, as mentioned in the introduction there are to the best of our knowledge no studies that have investigated this step-by step in men and women. That is also why we think that this study is very interesting for helping athletes and coaches as this process of detection would be automated. So it could be used in training to get a better understanding and faster learning for enhancement of performance.

I suggest you include more considerations about the limitation of your study.  Please add also which are the possible future lines.

We have rewritten the limitations part a little bit and added future lines in conclusion part.

CONCLUSIONS

The conclusion section should be further developed.Moreover, a practical implications section should be included. How these findings can help track and field coaches and practitioners? Which practical implications can it have in training sessions? 

We wanted to be short and concise. We have included a sentence with some practical implications. However, this is mostly speculation and has to be investigated further.

Round 2

Reviewer 1 Report

It was appreciated that the authors made some revisions to improve the manuscript. However, the problems with the clarity and presentation of study design and statistics did not resolve.

- The author agreed us that an ANOVA mixed design shall be used and they changed. However, it is not changed in the manuscript. It is currently read as “To compare the 50 m………..distance a 2 (sexes) x 13 (strides: repeated measures) ANOVA was used. In addition, a 2(fast-slow) x 13 stride: repeated measures) ANOVA between the faster and the slower ….was used”. I believe that it shall be written as “a two-way ANOVA mixed design (2 sexes x 13 strides: repeated measures)” and the latter shall be that the two-way ANOVA mixed designs were conducted twice for each of the sex strata.

- Line 151, “eight per group for men and five per group for women” could not be understood and why? How do the authors cut-off the faster and slower sprinters and why ?

- The authors gave up to conduct a 3-way ANOVA and I agree with their decision. However, their results appeared to have the 3-way interaction (i.e. significant difference in male x fast/slow, but not in female). Based on the study design, the authors shall consider the 3-way interaction effects, perhaps using other statistical methods, such as ANCOVA or the general linear model. In fact, it is not appropriate to cut-off fast/slower sprinters. The factor shall be treated as a continuous covariate factor in the study design.

- In my last comment, I mentioned that a correlation test is not used to make a distinction between groups. The authors moved that argument into the ANOVAs, which is equally incorrect. The authors shall be aware of the theory of the statistical test and the definition of the p-value.

- In my last comment, I emphasized the importance of considering interaction, main effect and simple main effect in the use of any 2-way tests. The authors simply added the interaction statistics in the results, which reflected their ignorance on how should conduct the statistics correctly. Depending on whether the interaction was of significant, there were different strategies to analyze the data, which shall be mentioned in the methods and correctly implemented in the results. For example, the authors mentioned that, in the first sentence of the results, “the sprint times were significantly faster in men in comparison to women”. Where was the “stride: repeated measures” in the test? In addition, the time was shorter, the time cannot be “faster”. Different sorts of these problems appeared throughout the results section.

- The definition of the time and stride was not clear though addressed previously. For example, what is the role of the stride on the total sprint time? That does not make sense. Picking the independent factor on any of the strides will not affect the total sprint time of that observation. Another example was about the stride in Figure 2. I don’t understand why 2 strides have a shorter time than one stride. It is supposed that was an accumulative relationship on the x-axis.

- the typo “sexe” appeared frequently.

Given that the critical aspect of data analysis was not seem correct and was not considerably improved during the review process, I do not recommend the publication of the article. The authors are recommended to consult a biostatistician for the correct presentation and implementation of the analysis.

Author Response

Comments to reviewer 1.

It was appreciated that the authors made some revisions to improve the manuscript. However, the problems with the clarity and presentation of study design and statistics did not resolve.

- The author agreed us that an ANOVA mixed design shall be used and they changed. However, it is not changed in the manuscript. It is currently read as “To compare the 50 m………..distance a 2 (sexes) x 13 (strides: repeated measures) ANOVA was used. In addition, a 2(fast-slow) x 13 stride: repeated measures) ANOVA between the faster and the slower ….was used”. I believe that it shall be written as “a two-way ANOVA mixed design (2 sexes x 13 strides: repeated measures)” and the latter shall be that the two-way ANOVA mixed designs were conducted twice for each of the sex strata.

We have changed it now according to the comments of the reviewer.

- Line 151, “eight per group for men and five per group for women” could not be understood and why? How do the authors cut-off the faster and slower sprinters and why?

As mentioned before the main purpose was to investigate the relationship between foot / ankle step-by-step kinematics measured by IMUs and sprint performance during a 50 m sprint in experienced male and female sprinters. This information could perhaps help trainers and athletes to make their training more efficient by the possible feedback they could give, if there are significant medium/large correlations. We could do it for the average strides, but perhaps the kinematics and the relationship with sprint times changes step-by-step. Thereby differences in step kinematics are different over the strides during the 50m which could distinguish faster and slower sprinters. The cut off was based upon the sprint times. That we also analysed kinematics besides the correlations, was to show if the correlations could easily show the differences between the faster compared to the slower sprinters, since that is easier to understand for most readers. That we choose the 5 and eight fastest and 5 and 8 slowest, without the subject in between was to have equal group sizes and this was type of analysis was also done in earlier studies. Since we only had 17 and 11 athletes we cut it in half with the subject in between not included. We hope that this reasoning is clear enough for the reviewer. We have included what fast sprinters mean (shorter 50m times) to indicate that it was based upon the 50m times.

- The authors gave up to conduct a 3-way ANOVA and I agree with their decision. However, their results appeared to have the 3-way interaction (i.e. significant difference in male x fast/slow, but not in female). Based on the study design, the authors shall consider the 3-way interaction effects, perhaps using other statistical methods, such as ANCOVA or the general linear model. In fact, it is not appropriate to cut-off fast/slower sprinters. The factor shall be treated as a continuous covariate factor in the study design.

As said before the main purpose was the relationships between foot kinematics and sprint performance, not the fast and slow sprinters. If we should have done a 3way ANOVA we would get the same results and no different interaction effects, since when you look at figure 2, 3 and 4 the development over the different strides are almost the same for fast and slow sprinters per sex, while it did not reach significance for women. This was caused by the low n in women as we stated as a limitation. As said before the comparison fast-slow sprinter was just to explain the relationships better for the reader. If the reviewer wants, we can delete the analysis between the fast and slower sprinters, but we think that this comparison adds clarity to the article about where the difference in kinematics occur during the sprint. So that the sprinters and athletes know where they have to focus upon, to enhance foot and sprint performance. We hope that the reviewer agrees with our point of view about this.

- In my last comment, I mentioned that a correlation test is not used to make a distinction between groups. The authors moved that argument into the ANOVAs, which is equally incorrect. The authors shall be aware of the theory of the statistical test and the definition of the p-value.

There are different ways to analyze data upon to make it clear for the reader we have decided for this types of analyses since correlations can only investigate the relationship between two variables it does not distinguish between directly between faster and slower sprinters, but just gives an indication over when plantar flexion velocity is higher it will result in fast times and at some steps it is more important than in other steps. By also analyzing it by dividing it in a faster and slower group (times) it is possible to show it in another way, which is for some better to understand than correlation. Strictly spoken it is enough by just showing the correlations since they already indicate if by higher angular velocity this would result in fast sprint times.  

- In my last comment, I emphasized the importance of considering interaction, main effect and simple main effect in the use of any 2-way tests. The authors simply added the interaction statistics in the results, which reflected their ignorance on how should conduct the statistics correctly. Depending on whether the interaction was of significant, there were different strategies to analyze the data, which shall be mentioned in the methods and correctly implemented in the results. For example, the authors mentioned that, in the first sentence of the results, “the sprint times were significantly faster in men in comparison to women”. Where was the “stride: repeated measures” in the test? In addition, the time was shorter, the time cannot be “faster”. Different sorts of these problems appeared throughout the results section.

We are fully aware of what interaction and main effects are when using a 2-way ANOVA since one of the authors is responsible for statistics courses here at university at bachelor, master and PhD level. As written in the version before we performed a 2-way ANOVA, but wrote it differently from what the reviewer expected. This is changed now to avoid confusion. That we earlier did not mention the possible interaction effects was due to the reason that we did not find any significant interaction effects and therefore chose not to mention them. However, that was perhaps not so smart. That is why they were included in the previous revision. However, it does not change the findings. The first 13 strides over the 50m sprints were analysed between sex since all had at least 13 strides. Since every person had 13 strides and changes between each strid can occur a repeated measures design on 13 strides was performed. When you perform a comparison between so many strides you could chose for a Bonnferoni correction. However, that is a very conservative correction, and we were only interested in the development from step to step. Therefore, the LSD post hoc compassion. We understand the confusion about faster and shorter, which we now have defined in the methods part. Faster sprinters had a shorter 50m sprint time. We have also changed fast sprint times in shorter to avoid confusion.

- The definition of the time and stride was not clear though addressed previously. For example, what is the role of the stride on the total sprint time? That does not make sense. Picking the independent factor on any of the strides will not affect the total sprint time of that observation. Another example was about the stride in Figure 2. I don’t understand why 2 strides have a shorter time than one stride. It is supposed that was an accumulative relationship on the x-axis.

We think that the reviewer misunderstood the definition of time, timing and strides. We have measured all 3 events for each step with the timing between each event for each step. This was averaged for each stride (left and right step). This was all mentioned in the methods part. The time between each event changed over the different strides for the maximal dorsiflexion velocity to toe off, which was shown in figure 2, 3 and 4. However, we mentioned in the text that it significantly changed. This was not indicated in the figures to avoid too many extra lines or signs, which would make it difficult for the reader to understand. So in the first strides of the 50m print the time between the 2 events was much longer than at stride 6-7 since they were at maximal velocity. Since we did it for each stride the comparison between fast and slower sprinters was easier to indicate where the differences occur (from stride 8 and onwards), which is important information for the reader.  

- the typo “sexe” appeared frequently.

This is changed in sex now.

Given that the critical aspect of data analysis was not seem correct and was not considerably improved during the review process, I do not recommend the publication of the article. The authors are recommended to consult a biostatistician for the correct presentation and implementation of the analysis.

Hereby the authors feel a bit offended, since one of the authors is responsible to statistics courses at bachelor, master and PhD level at university and has performed many studies in which similar statistics were used, without questioning the statistics, which are correctly performed, just not everything was mentioned in the first version. Furthermore, it is possible to do more difficult statistical analysis, which makes it difficult for the reader to understand, but this will not change the findings and conclusion. We hope that the reviewer agrees about that.